# Associations between inflexible job conditions, health and healthcare utilisation in England: retrospective cross-sectional study

Charlie Moss ![ORCID],[1] Luke Aaron Munford ![ORCID],[1] Matt Sutton[1,2]

[1]Health Organisation, Policy and Economics (HOPE), Centre for Primary Care and Health Services Research, School of Health Sciences, The University of Manchester, Manchester, UK
[2]Melbourne Institute: Applied Economic and Social Research, The University of Melbourne, Melbourne, Victoria, Australia

**Correspondence to**
Charlie Moss;
charlie.moss@manchester.ac.uk

## ABSTRACT

**Objectives** To estimate the strength of association between having an inflexible job and health-related quality of life and healthcare utilisation; and to explore heterogeneity in the effects by gender, age and area-level deprivation.
**Design** Retrospective cross-sectional study.
**Setting** Seven waves of the English General Practice Patient Survey between 2012 and 2017.
**Participants** 1 232 884 people aged 16–64 years and in full-time employment. We measured job inflexibility by inability to take time away from work during usual working hours to seek medical care.
**Primary and secondary outcome measures** Health-related quality of life (EQ-5D-5L); number of months since the respondent last saw a general practitioner (GP) or nurse; use of out-of-hours general practice in the past 6 months. We used regression analyses to estimate the strength of association between outcomes and having an inflexible job, adjusting for person and area-level characteristics.
**Results** One-third of respondents reported job inflexibility. The probability of job inflexibility was higher at younger ages and in more deprived areas. Job inflexibility was associated with lower EQ-5D-5L utility scores of 0.017 (95% CI 0.016 to 0.018) for women and 0.016 (95% CI 0.015 to 0.017) for men. Women were more affected than men in the mental health domain. The reduction in health-related quality of life associated with having an inflexible job was greater for employees who were older or lived in more deprived areas. Having an inflexible job was associated with a longer time since the last visit to their GP of 0.234 (95% CI 0.201 to 0.268) months for women and 0.199 (95% CI 0.152 to 0.183) months for men.
**Conclusions** Inequalities in the prevalence of inflexible jobs contribute to inequalities in health. One mechanism may be through reduced access to healthcare. Policymakers and employers should ensure that all employees have sufficient job flexibility to protect their health.

## INTRODUCTION

Employee-oriented work flexibility refers to workers' discretion over some aspect of their job, such as location or hours of work. Examples include 'flexitime', involving discretion

## STRENGTHS AND LIMITATIONS OF THIS STUDY

⇒ Large sample of 1 232 884 survey respondents.
⇒ We were able to use the EQ-5D-5L instrument which is a validated measure of health-related quality of life.
⇒ The General Practice Patient Survey is a repeated cross-sectional survey. This makes it difficult to establish causality.
⇒ The measure of job flexibility may not fully capture other time constraints that are unrelated to healthcare utilisation during working hours.

in start and finish times, and the option to work from home. It is intended to improve working and social life for the employee by increasing time sovereignty.[1]

The relationship between work flexibility and health is an increasingly salient consideration due to changes in working norms induced by the coronavirus pandemic.[2–6] There is new impetus to understand the mechanisms via which any health benefits of flexible working occur, and for whom.

Beckers *et al* provide a summary of the motivational and occupational health theories which give rise to the assumed benefits of increased worktime control.[7] Several are relevant for considering the effects that flexibility may have on employees' health. Effort–Recovery Theory[8] suggests that a flexible job may allow employees to improve their recovery after exerting effort at work, for example, by sleeping at times most optimal for them. Improved short-term recovery is likely to benefit physical and mental health in the long term.[9 10]

Through the lens of theories of work–home interaction,[11] flexibility allows employees to better meet obligations, needs and activities in their private lives. This is particularly relevant for health, where many activities that employees undertake outside of work can directly affect health. An inflexible job may

limit an individual's ability to engage in health-promoting activities such as accessing healthcare, exercising and cooking fresh food.[12]

Overall, the applied literature suggests that work flexibility may be positively associated with health and negatively associated with outcomes such as burnout, fatigue, perceived stress and sickness absence.[1 13–22] A systematic review of the effect of flexibility on mental health concluded that flexibility had small positive effects on outcomes relating to mental health.[23]

However, previous studies suffer from several limitations. Most likely due to insufficient sample size, evidence of variation between population subgroups is lacking, even though we may expect some groups of employees to gain more from work flexibility (older employees and employees with other responsibilities).[24] Previous studies are also based predominantly on work-focused surveys that tend to include only crude measures of health and no information on healthcare utilisation.

We were unable to find a study of the effect of work flexibility on healthcare utilisation, despite this being a potentially important pathway contributing to the overall effect of work flexibility on health. For employees working 'standard' daytime hours, those are the same hours that most routine healthcare provision is available. Although there may be some provision outside of 'standard' hours, employees may be more likely to forego healthcare appointments if their job does not afford them the flexibility to attend during working hours.

We use large, repeated, health-focused cross-sectional surveys for England to consider full-time employed individuals who report their jobs are so inflexible that they cannot attend primary care appointments during working hours. We map the geographical distribution of inflexible jobs, estimate the associations between job flexibility and health and healthcare utilisation, and exploit the large sample for subgroup analyses by gender, age and deprivation.

## METHODS

### Study population

We used data from the national General Practice Patient Survey (GPPS) in England.[25] The survey is distributed by post to over 2 million people annually. It is sent to a sample of patients registered at each general practice in England and includes questions on job flexibility, health, primary healthcare use, socioeconomic characteristics and the EQ-5D measure of health-related quality of life (HRQoL). We used data from six survey years, 2012–2017, when data on HRQoL were collected consistently using the five-level version of the EQ-5D (EQ-5D-5L).[26]

### Procedures

If the respondent reported being in paid employment, they were asked: 'If you need to see a general practitioner (GP) at your GP surgery during your typical working hours, can you take time away from your work to do this? Yes/No'.

The EQ-5D has five domains: 'mobility', 'self-care', 'usual activities', 'pain/discomfort' and 'anxiety/depression'. Individuals were asked to report their health today in each domain on a 5-point ordinal scale, where (1) corresponds to no problems; (2) slight problems; (3) moderate problems; (4) severe problems and (5) extreme problems/unable to do activities. We analysed each domain separately and a composite utility score, obtained by mapping the five-level to the three-level version using the cross-walk tool described in van Hout *et al*.[27]

We also analysed utilisation of three types of healthcare services. The first two types were GP and nurse visits. The GPPS asks how recently the respondent has seen or spoken to (1) a GP and (2) a nurse at their GP practice, with five possible responses ranging from 'in the past 3 months' to 'I have never seen a (GP/nurse) from my surgery'. We recoded these responses to create intervals for the number of months since the respondent had seen a GP/nurse (see online supplemental table 1). The third type of healthcare was use of out-of-hours primary care. Up until 2015, the GPPS asked the question 'In the past 6 months, have you tried to call an out-of-hours GP service when [your usual] surgery was closed? Yes, for myself/ Yes, for someone else/No'. We considered only calls made for the respondent themself.

The GPPS contains information on the gender of each respondent, their age (in age bands: 18–24 years; 25–34 years; 35–44 years; 45–54 years; 55–64 years), their ethnicity (white; mixed; Asian; black; other), whether they had ever smoked, and whether they were a parent/guardian to a child aged 16 years and under, and whether they had a long-term condition.

The GPPS asks 'last time you wanted to see or speak to a nurse from your GP surgery, were you able to get an appointment to see or speak to someone?'. We used the percentage of respondents who replied 'yes' at each practice as an indicator of appointment availability at the practice level.

England is divided into 326 geographical areas called local authority districts (LADs), and is further broken down into a collection of 32 844 smaller geographical areas called lower-level super output areas (LSOAs). LSOAs contain on average 1500 residents. Levels of deprivation were calculated at LSOA level using the 2015 iteration of the Index of Multiple Deprivation.[28] We used deciles of area deprivation, where 1 is the least deprived decile and 10 is the most deprived decile.

We restricted the main analysis to individuals who reported working full-time. We did this to avoid issues associated with part-time working. For example, we believe that part-time and full-time individuals face different choice sets when it comes to time allocation. As stated above, the main variable we are concerned with relates to an individual's ability to visit a GP during their *usual working hours*. This definition of usual working hours is likely to vary between full-time and part-time employees.

As a robustness check, we relaxed the assumption relating to full-time employment by including part-time workers in the estimation sample.

We also restricted the analysis to those within the statutory working-age range, 18–64 years. This is to minimise endogeneity generated by voluntary participation in work beyond state retirement age.

We retained observations with non-missing data for all outcomes and covariates. In the online supplemental material, we provide a flow chart which shows the number of observations dropped at each stage of data preparation (online supplemental figure 1).

### Statistical analyses

We calculated the percentage of GPPS respondents who reported having an inflexible job in each of the 326 LADs that existed as of 2017 in England. We used spmap[29] to create a heat map showing the percentage for each area. We compared this with a heat map of the mean EQ-5D utility score for GPPS respondents in each area.

We used linear regression to estimate the strength of the overall association between the EQ-5D composite utility score and having an inflexible job, adjusting for person characteristics, LSOA characteristics, and fixed effects for local authorities and survey years. We used ordered logistic regression to analyse the EQ-5D domains.

We used interval regression[30] to estimate the strength of association between the number of months since the respondent last saw (1) a GP or (2) a nurse and having an inflexible job; and logistic regression for whether the respondent had tried to access out-of-hours primary care in the past 6 months. We included the same covariates in these regressions in addition to: indicators for specific long-term conditions, and an indicator of appointment availability at the practice level.

The main analyses do not combine the associations of job inflexibility with health and with healthcare utilisation. As shown in figure 1, it is possible that healthcare utilisation is one of the reasons health is adversely affected. If having an inflexible job acts as a barrier to accessing healthcare when it is needed, this would result in a negative effect on healthcare use. Because healthcare is expected to have a positive effect on health, this would create an additional negative effect of an inflexible job on health.

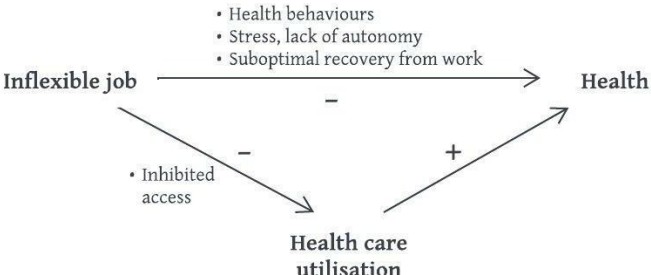

**Figure 1** Graph of the hypothesised effect of an inflexible job on health.

To test this pathway, we used two-stage least squares to decompose the effect of healthcare utilisation on health into that which is related to (1) having an inflexible job and (2) all other factors that determine healthcare utilisation. In the first stage, we estimated the association between the time since the respondent last saw a GP or nurse, and whether they had an inflexible job. In the second stage, we estimated the association between the predicted values for the time since the respondent last saw a GP/nurse and HRQoL. In these regressions, we treated the time since the respondent last saw a GP/nurse as a continuous variable ranging from one to five (online supplemental table 1).

We present the equations for each regression specification as online supplemental material.

We estimated all models with robust SEs, and used the weights provided with the survey data which correct for the GPPS sampling design and reduce the impact of non-response bias.

To investigate differences between subgroups, we estimated all models for men and women separately. We additionally estimated the effect of having an inflexible job on HRQoL with interactions between having an inflexible job and: gender, area deprivation decile and age group.

### Patient and public involvement

Patients or the public were not involved in the design of this study.

### RESULTS

The estimation sample comprised 570 626 women and 662 258 men, aged 18–64 years and in full-time employment. Thirty-two per cent of respondents reported that their job was inflexible. There were gender differences, with men reporting lower levels than women (29% compared with 35%; table 1).

Panel (a) of online supplemental figure 2 shows the geographical distribution of inflexible jobs. The areas with the highest concentration of inflexible jobs (darkest shading) were areas of the north-east, the north-west, west Midlands and London. These areas are traditionally the areas with the highest levels of deprivation and the highest concentration of manual jobs. Online supplemental tables 2 and 3 provide greater detail on which local authorities had the highest and lowest percentages of respondents who reported inflexible jobs.

Panel (b) of online supplemental figure 2 shows the geographical distribution of mean EQ-5D utility score. Darker areas indicate worse health, and these are present in both the north and the south. London has a high concentration of light-coloured areas which represent good average health. Excluding London, there was an apparent correlation between the two measures: areas with higher rates of inflexible jobs, on average, had lower levels of health. The raw individual level correlation of −0.051 between reporting an inflexible job and EQ-5D utility score confirms this.

| Table 1 | Selected summary statistics for men and women | |
|---|---|---|
| | **Female** | **Male** |
| Outcomes | | |
| Mean EQ-5D utility score (SD) | 0.89 (0.14) | 0.89 (0.14) |
| Months since last saw GP (%) | | |
| 0–3 | 53.8 | 38.4 |
| 3–6 | 19.2 | 18.2 |
| 6–12 | 14.6 | 18.0 |
| 12–24 | 11.3 | 22.8 |
| 24+ | 1.1 | 2.7 |
| Months since last saw nurse (%) | | |
| 0–3 | 33.8 | 21.1 |
| 3–6 | 18.9 | 13.4 |
| 6–12 | 19.1 | 15.7 |
| 12–24 | 24.0 | 35.9 |
| 24+ | 4.2 | 13.8 |
| Out of hours (% contacted in last 6 months) | 3.2 | 2.2 |
| Key variable (%) | | |
| Inflexible job | 35.4 | 28.9 |
| Individual-level characteristics (%) | | |
| Parent | 24.7 | 34.9 |
| Aged 18–24 years | 11.0 | 7.6 |
| Aged 25–34 years | 28.5 | 23.9 |
| Aged 35–44 years | 22.6 | 25.9 |
| Aged 45–54 years | 25.9 | 27.1 |
| Aged 55–64 years | 12.1 | 15.5 |
| Ethnicity: white | 88.8 | 86.7 |
| Ethnicity: mixed | 1.3 | 1.0 |
| Ethnicity: Asian | 5.5 | 7.5 |
| Ethnicity: black | 2.8 | 2.4 |
| Ethnicity: other | 1.6 | 2.3 |
| Never smoked | 62.7 | 56.4 |
| General practice-level covariates | | |
| Mean appointment availability (SD) | 90.2 (6.0) | 90.2 (6.1) |
| | N=570 626 | N=662 258 |

Data source: GPPS pooled data for 2012–2017; respondents in full-time employment.
Weighted using GPPS weights.
GP, general practitioner; GPPS, General Practice Patient Survey.

The mean EQ-5D utility score was 0.89, for both women and men. The domain with the highest percentage of respondents reporting 'no problems' was self-care (98%; online supplemental table 4). In descending order, the percentage of respondents reporting 'no problems' in other domains were: mobility (91%); usual activities (88%); anxiety/depression (74%); and pain/discomfort (64%). All domains were very similar across gender, except anxiety/depression, where 77% of men reported 'no problems' compared with 71% of women.

Forty-one per cent of respondents reported that they have a long-term condition. The most prevalent long-term condition among men was high blood pressure (10%); 'another long-term condition' was the most prevalent for women (10%; online supplemental table 5).

Thirty-five per cent of men and 25% of women were parents, and the modal age group was 45–54 years of age. The sample was predominantly white (88%), with 7% Asian, 3% black, 2% other and 1% mixed (table 1).

Individuals in the sample tended to have seen a GP more recently than a nurse (table 1). Forty-four per cent saw a GP in the last 6 months, compared with 26% who saw a nurse. Two per cent of individuals had never seen a GP in their practice, compared with 10% who had never seen a nurse.

Respondents living in more deprived areas were more likely to have an inflexible job compared with those living in less deprived areas (table 2, columns (1) and (2)). Older respondents were less likely than younger respondents to report having an inflexible job. Women were more likely to have an inflexible job than men (online supplemental table 6, column (1)).

Women whose jobs were inflexible had average EQ-5D utility scores that were 0.017 (95% CI 0.016 to 0.018) lower than their counterparts with a flexible job (table 2, column (3)). Men whose jobs were inflexible had average EQ-5D utility scores that were 0.016 (95% CI 0.015 to 0.017) lower than their counterparts with a flexible job (table 2, column (4)). The SD of the EQ-5D utility score was 0.14 for men and women (table 1). This means that the reduction was about 12% of one SD for women and 11% for men.

These effects remain when part-time workers are included in the estimation sample (online supplemental table 7). The main coefficients of interest (on 'inflexible job') are almost identical in magnitude and statistical significance.

For the five separate domains of the EQ-5D, having an inflexible job was associated with a lower probability of reporting 'no problems', and a greater probability of all other responses (table 3). The magnitude of the effects on the probability of reporting 'no problems' were slightly larger in absolute magnitude for men in the domains of mobility, self-care, usual activities and pain/discomfort. However, for the anxiety/depression domain, the effect was larger for women than men.

There was both a deprivation and age gradient on the effect of an inflexible job on health (figure 2 and online supplemental table 6, column (2)). The interaction effects suggest that the negative effect of having an inflexible job on health was larger in magnitude for people living in a more deprived area compared with those living in a less deprived area; larger in magnitude for older people compared with younger people; and very slightly larger for women, compared with men.

**Table 2** The effect of covariates on reporting an inflexible job and the effect of an inflexible job on the EQ-5D utility score

| | Inflexible job | | EQ-5D utility score | |
| --- | --- | --- | --- | --- |
| | Female | Male | Female | Male |
| | (1) | (2) | (3) | (4) |
| Inflexible job | | | −0.017 | −0.016 |
| | | | (−0.018 to −0.016) | (−0.017 to −0.015) |
| Parent | −0.005 | 0.002 | 0.018 | 0.011 |
| | (−0.008 to −0.001) | (−0.002 to 0.005) | (0.017 to 0.019) | (0.010 to 0.012) |
| Aged 25–34 years* | −0.064 | −0.061 | 0.001 | −0.019 |
| | (−0.071 to −0.057) | (−0.070 to −0.053) | (−0.001 to 0.003) | (−0.021 to −0.017) |
| Aged 35–44 years* | −0.097 | −0.089 | −0.019 | −0.038 |
| | (−0.104 to −0.090) | (−0.098 to −0.081) | (−0.021 to −0.017) | (−0.040 to −0.036) |
| Aged 45–54 years* | −0.106 | −0.101 | −0.038 | −0.056 |
| | (−0.113 to −0.099) | (−0.109 to −0.093) | (−0.040 to −0.036) | (−0.058 to −0.054) |
| Aged 55–64 years* | −0.134 | −0.131 | −0.050 | −0.071 |
| | (−0.141 to −0.127) | (−0.139 to −0.124) | (−0.052 to −0.048) | (−0.073 to −0.069) |
| Ethnicity: mixed† | 0.002 | 0.001 | −0.010 | −0.004 |
| | (−0.012 to 0.016) | (−0.014 to 0.017) | (−0.014 to −0.005) | (−0.009 to 0.000) |
| Ethnicity: Asian† | 0.006 | 0.022 | 0.005 | 0.008 |
| | (−0.001 to 0.013) | (0.016 to 0.028) | (0.003 to 0.007) | (0.006 to 0.009) |
| Ethnicity: black† | −0.034 | −0.011 | 0.001 | 0.017 |
| | (−0.043 to −0.025) | (−0.020 to −0.002) | (−0.002 to 0.003) | (0.015 to 0.020) |
| Ethnicity: other† | −0.000 | 0.013 | −0.010 | −0.004 |
| | (−0.012 to 0.012) | (0.003 to 0.023) | (−0.014 to −0.006) | (−0.008 to −0.001) |
| Never smoked | 0.012 | −0.024 | 0.026 | 0.025 |
| | (0.009 to 0.016) | (−0.027 to −0.021) | (0.025 to 0.027) | (0.024 to 0.026) |
| Deprivation decile 1‡ | −0.007 | −0.031 | 0.009 | 0.011 |
| (least deprived) | (−0.014 to 0.000) | (−0.038 to −0.025) | (0.007 to 0.011) | (0.009 to 0.013) |
| Deprivation decile 2‡ | −0.012 | −0.019 | 0.006 | 0.008 |
| | (−0.019 to −0.005) | (−0.026 to −0.013) | (0.004 to 0.008) | (0.006 to 0.010) |
| Deprivation decile 3‡ | −0.010 | −0.018 | 0.004 | 0.006 |
| | (−0.017 to −0.003) | (−0.024 to −0.011) | (0.002 to 0.006) | (0.004 to 0.007) |
| Deprivation decile 4‡ | −0.005 | −0.004 | 0.002 | 0.003 |
| | (−0.012 to 0.002) | (−0.011 to 0.002) | (−0.000 to 0.003) | (0.001 to 0.005) |
| Deprivation decile 6‡ | 0.004 | 0.006 | −0.005 | −0.003 |
| | (−0.003 to 0.011) | (−0.000 to 0.013) | (−0.007 to −0.003) | (−0.005 to −0.001) |
| Deprivation decile 7‡ | 0.010 | 0.016 | −0.010 | −0.008 |
| | (0.003 to 0.017) | (0.010 to 0.023) | (−0.012 to −0.008) | (−0.010 to −0.006) |
| Deprivation decile 8‡ | 0.015 | 0.024 | −0.012 | −0.010 |
| | (0.008 to 0.022) | (0.018 to 0.031) | (−0.014 to −0.010) | (−0.012 to −0.008) |
| Deprivation decile 9‡ | 0.029 | 0.041 | −0.016 | −0.013 |
| | (0.022 to 0.036) | (0.034 to 0.048) | (−0.018 to −0.014) | (−0.015 to −0.011) |
| Deprivation decile 10‡ | 0.042 | 0.053 | −0.023 | −0.018 |
| (most deprived) | (0.035 to 0.050) | (0.046 to 0.061) | (−0.025 to −0.021) | (−0.020 to −0.015) |
| N | 570 626 | 662 258 | 570 626 | 662 258 |
| Adjusted $R^2$ | 0.013 | 0.018 | 0.040 | 0.044 |

**Table 2** Continued

| | Inflexible job | | EQ-5D utility score | |
|---|---|---|---|---|
| | **Female** | **Male** | **Female** | **Male** |
| | (1) | (2) | (3) | (4) |

Data source: GPPS pooled data for 2012–2017.
95% CIs in parentheses.
Estimated using linear regression.
Models additionally include fixed effects for survey year and local authority district of residence.
*The reference category for age group is 16–24 years.
†The reference category for ethnic group is white.
‡The reference category for deprivation is decile 5.
GPPS, General Practice Patient Survey.

Having an inflexible job was associated with less recent utilisation of both GPs and nurses (table 4 and online supplemental table 8). For women with an inflexible job, it had been an additional 0.234 (95% CI 0.201 to 0.268) months since they had seen a GP at their surgery, compared with women with a flexible job. The magnitude of this effect was greater for nurse appointments than GP appointments; for women with an inflexible job, it had been an additional 0.569 (95% CI 0.519 to 0.618) months since they had seen a nurse at their GP surgery, compared with women with a flexible job. For men with an inflexible job, it had been an additional 0.199 (95% CI 0.152 to 0.247) months since they last saw a GP at their surgery, and an additional 0.653 (95% CI 0.586 to 0.720) months since they last saw a nurse at their GP surgery, compared with men with a flexible job. These results are robust to a range of alternative definitions of each interval in the interval regressions (online supplemental tables 9 and 10).

Conversely, individuals with inflexible jobs made more use of 'out-of-hours' GP appointments than their counterparts with flexible jobs (table 4 and online supplemental table 11).

The results of the second stage of the two-stage least squares regressions show the effect of healthcare use on the EQ-5D utility score when healthcare use is predicted by job inflexibility (table 5 and online supplemental table 12). A longer predicted duration of time since the individual had seen a GP (or nurse) was negatively associated with the EQ-5D utility score. Therefore, the element of healthcare use that was negatively associated with job inflexibility was also associated with lower health.

**Table 3** The average marginal effect of having an inflexible job on the probability of reporting each EQ-5D domain level

| | Mobility | | Self-care | | Usual activities | | Pain/discomfort | | Anxiety/depression | |
|---|---|---|---|---|---|---|---|---|---|---|
| | **Female** | **Male** | **Female** | **Male** | **Female** | **Male** | **Female** | **Male** | **Female** | **Male** |
| No problems | −0.014 | −0.017 | −0.002 | −0.003 | −0.023 | −0.025 | −0.051 | −0.052 | −0.035 | −0.021 |
| | (−0.015 to −0.012) | (−0.019 to −0.015) | (−0.003 to −0.002) | (−0.004 to −0.002) | (−0.026 to −0.021) | (−0.027 to −0.023) | (−0.054 to −0.048) | (−0.055 to −0.049) | (−0.038 to −0.032) | (−0.024 to −0.018) |
| Slight problems | 0.010 | 0.012 | 0.002 | 0.002 | 0.017 | 0.018 | 0.030 | 0.033 | 0.020 | 0.014 |
| | (0.008 to 0.011) | (0.011 to 0.013) | (0.001 to 0.002) | (0.002 to 0.003) | (0.016 to 0.019) | (0.016 to 0.019) | (0.028 to 0.032) | (0.031 to 0.035) | (0.019 to 0.023) | (0.012 to 0.015) |
| Moderate problems | 0.003 | 0.004 | 0.001 | 0.001 | 0.005 | 0.005 | 0.017 | 0.016 | 0.011 | 0.006 |
| | (0.003 to 0.003) | (0.003 to 0.004) | (0.0004 to 0.0007) | (0.0005 to 0.0008) | (0.005 to 0.006) | (0.005 to 0.006) | (0.016 to 0.018) | (0.015 to 0.017) | (0.010 to 0.012) | (0.005 to 0.007) |
| Severe problems | 0.001 | 0.001 | 0.0001 | 0.0001 | 0.001 | 0.001 | 0.004 | 0.003 | 0.002 | 0.001 |
| | (0.001 to 0.001) | (0.001 to 0.001) | (0.0001 to 0.0001) | (0.0001 to 0.0001) | (0.001 to 0.001) | (0.001 to 0.001) | (0.004 to 0.004) | (0.003 to 0.003) | (0.002 to 0.002) | (0.001 to 0.001) |
| Unable to/extreme problems | 0.0002 | 0.0003 | 0.0001 | 0.0001 | 0.0004 | 0.0004 | 0.001 | 0.0005 | 0.001 | 0.0003 |
| | (0.0002 to 0.0002) | (0.0002 to 0.0003) | (0.0001 to 0.0001) | (0.0001 to 0.0002) | (0.0004 to 0.0005) | (0.004 to 0.0005) | (0.001 to 0.001) | (0.0004 to 0.0005) | (0.0005 to 0.001) | (0.0003 to 0.0004) |
| N | 570 626 | 662 258 | 570 626 | 662 258 | 570 626 | 662 258 | 570 626 | 662 258 | 570 626 | 662 258 |
| Pseudo R² | 0.044 | 0.045 | 0.033 | 0.033 | 0.023 | 0.029 | 0.028 | 0.031 | 0.012 | 0.011 |

Data source: GPPS pooled data for 2012–2017.
95% CIs in parentheses.
Models additionally include indicators for: parental status, 10-year age group, ethnicity, whether the respondent has ever smoked, Index of Multiple Deprivation decile, and fixed effects for survey year and local authority district of residence.
Estimated using ordered logistic regression.
GPPS, General Practice Patient Survey.

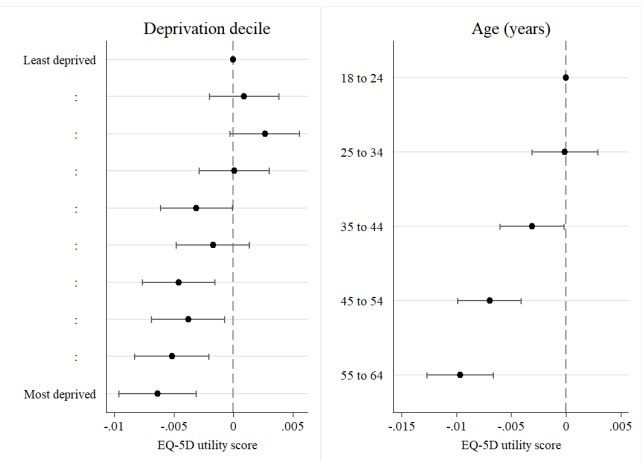

**Figure 2** Subgroup effects: the interaction between having an inflexible job and deprivation decile (left) and age (right). Data source: GPPS pooled data for 2012–2017; 1 232 884 respondents in full-time employment. Markers show point estimates of interaction terms; bars show 95% CIs. Model estimated using linear regression. Model additionally includes indicators for: gender, gender×inflexible job, parental status, 10-year age group, ethnicity, whether the respondent has ever smoked, Index of Multiple Deprivation decile, and fixed effects for survey year and local authority district of residence. GPPS, General Practice Patient Survey.

## DISCUSSION

We found that individuals who had inflexible jobs were less likely to be in good health and the result was common to men and women. Our subgroup analyses of different age groups and people living in areas of differing deprivation suggest that the most affected groups were older people and those living in the most deprived areas. Older people were less likely to experience inflexible jobs, but

their health was more affected if they did, compared with younger people. People living in more deprived areas were both more likely to have an inflexible job, and their health was more affected than those in less deprived areas.

When considering subdomains of health separately, we found that women were more affected in the anxiety/depression domain than men. This is a similar finding to Moen *et al*[13] which found that women especially benefited from a reduction in perceived stress and psychological stress when randomised into an intervention with greater worktime control.

We found that individuals with inflexible jobs made fewer visits to see their GP or a nurse compared with individuals with flexible jobs. As would perhaps be expected, individuals with inflexible jobs used out-of-hours services more often than individuals with flexible jobs, but still overall went to see their GP/nurse less frequently. It is possible that the lower opportunity cost of time for workers who were able to consume healthcare during working hours contributed to their more frequent visits to their GP.

Typically, the onus has been put on the healthcare system to accommodate inflexible jobs by offering 'out-of-hours' appointments in the evenings and on weekends, at greater cost to the National Health Service than regular GP appointments. However, perhaps there should be a greater responsibility for employers to ensure that systems are in place so that their employees are able to attend routine healthcare appointments during usual working hours, when they need to.

We add to the evidence base on the relationship between work flexibility and health, and to the very limited evidence base on the relationship between work flexibility and healthcare utilisation. Our analyses suggest

**Table 4** The effects of an inflexible job on healthcare utilisation

| | Months since last saw GP | | Months since last saw nurse | | Tried to call out-of-hours service in past 6 months | |
|---|---|---|---|---|---|---|
| | **Female** | **Male** | **Female** | **Male** | **Female** | **Male** |
| | **(1)** | **(2)** | **(3)** | **(4)** | **(5)** | **(6)** |
| Inflexible job | 0.234 | 0.199 | 0.569 | 0.653 | 0.012 | 0.010 |
| | (0.201 to 0.268) | (0.152 to 0.247) | (0.519 to 0.618) | (0.586 to 0.720) | (0.010 to 0.015) | (0.008 to 0.012) |
| N | 570 626 | 662 258 | 570 626 | 662 258 | 293 266 | 336 890 |
| Pseudo R$^2$ | | | | | 0.028 | 0.260 |
| AIC | 2 203 151 | 3 576 803 | 2 317 874 | 3 453 027 | | |
| BIC | 2 207 315 | 3 581 022 | 2 322 038 | 3 457 246 | | |

Data source (1)–(4): GPPS pooled data for 2012–2017.
Data source (5) and (6): GPPS pooled data for 2012–2014.
95% CIs in parentheses.
All models additionally include indicators for: parental status, 10-year age group, ethnicity, whether the respondent has ever smoked, Index of Multiple Deprivation decile, 16 long-term conditions, practice-level appointment availability, and fixed effects for survey year and local authority district of residence.
(1)–(4) estimated using interval regression.
(5) and (6) estimated using logistic regression, results show average marginal effects.
AIC, Akaike information criterion; BIC, Bayesian information criterion; GP, general practitioner; GPPS, General Practice Patient Survey.

**Table 5** Two-stage least squares regression results: the mediating effect of GP visits in the effect of an inflexible job on health

| | First stage: time since last saw GP | | Second stage: EQ-5D utility score | |
| --- | --- | --- | --- | --- |
| | Female | Male | Female | Male |
| | (1) | (2) | (3) | (4) |
| Time since last saw GP | | | −0.355 | −0.539 |
| | | | (−0.419 to −0.291) | (−0.715 to −0.362) |
| Inflexible job | 0.047 | 0.029 | | |
| | (0.039 to 0.054) | (0.020 to 0.038) | | |
| Parent | −0.279 | −0.140 | −0.082 | −0.064 |
| | (−0.288 to −0.271) | (−0.149 to −0.130) | (−0.100 to −0.063) | (−0.090 to −0.039) |
| Aged 25–34 years* | 0.025 | −0.060 | 0.01 | −0.051 |
| | (0.010 to 0.041) | (−0.082 to −0.038) | (0.004 to 0.016) | (−0.068 to −0.035) |
| Aged 35–44 years* | 0.14 | −0.169 | 0.03 | −0.129 |
| | (0.124 to 0.156) | (−0.190 to −0.147) | (0.020 to 0.041) | (−0.161 to −0.096) |
| Aged 45–54 years* | 0.161 | −0.279 | 0.019 | −0.206 |
| | (0.146 to 0.175) | (−0.300 to −0.259) | (0.007 to 0.031) | (−0.258 to −0.155) |
| Aged 55–64 years* | 0.146 | −0.486 | 0.002 | −0.333 |
| | (0.130 to 0.161) | (−0.507 to −0.466) | (−0.009 to 0.013) | (−0.420 to −0.246) |
| Ethnicity: mixed† | −0.029 | −0.064 | −0.02 | −0.039 |
| | (−0.060 to 0.002) | (−0.106 to −0.022) | (−0.033 to −0.007) | (−0.065 to −0.012) |
| Ethnicity: Asian† | 0.014 | −0.227 | 0.01 | −0.115 |
| | (−0.002 to 0.031) | (−0.243 to −0.211) | (0.003 to 0.016) | (−0.156 to −0.074) |
| Ethnicity: black† | −0.11 | −0.182 | −0.038 | −0.080 |
| | (−0.129 to −0.091) | (−0.205 to −0.158) | (−0.049 to −0.028) | (−0.115 to −0.046) |
| Ethnicity: other† | −0.022 | −0.185 | −0.018 | −0.104 |
| | (−0.051 to 0.007) | (−0.212 to −0.159) | (−0.029 to −0.006) | (−0.140 to −0.068) |
| Never smoked | 0.114 | 0.101 | 0.067 | 0.080 |
| | (0.107 to 0.121) | (0.093 to 0.109) | (0.059 to 0.075) | (0.061 to 0.098) |
| Deprivation decile 1‡ | 0.007 | 0.027 | 0.012 | 0.025 |
| (least deprived) | (−0.010 to 0.023) | (0.008 to 0.045) | (0.005 to 0.018) | (0.014 to 0.037) |
| Deprivation decile 2‡ | −0.003 | 0.016 | 0.005 | 0.016 |
| | (−0.019 to 0.013) | (−0.002 to 0.034) | (−0.002 to 0.011) | (0.006 to 0.027) |
| Deprivation decile 3‡ | 0.008 | −0.000 | 0.007 | 0.005 |
| | (−0.008 to 0.024) | (−0.018 to 0.018) | (0.001 to 0.013) | (−0.005 to 0.016) |
| Deprivation decile 4‡ | −0.012 | 0.000 | −0.003 | 0.003 |
| | (−0.028 to 0.003) | (−0.018 to 0.018) | (−0.009 to 0.003) | (−0.007 to 0.014) |
| Deprivation decile 6‡ | 0.002 | −0.008 | −0.004 | −0.008 |
| | (−0.014 to 0.018) | (−0.026 to 0.010) | (−0.010 to 0.002) | (−0.018 to 0.003) |
| Deprivation decile 7‡ | −0.028 | −0.011 | −0.02 | −0.014 |
| | (−0.044 to −0.013) | (−0.030 to 0.007) | (−0.026 to −0.013) | (−0.025 to −0.003) |
| Deprivation decile 8‡ | −0.018 | −0.007 | −0.019 | −0.013 |
| | (−0.034 to −0.002) | (−0.025 to 0.012) | (−0.025 to −0.012) | (−0.024 to −0.003) |
| Deprivation decile 9‡ | −0.014 | −0.022 | −0.021 | −0.025 |
| | (−0.031 to 0.002) | (−0.041 to −0.003) | (−0.027 to −0.014) | (−0.036 to −0.013) |
| Deprivation decile 10‡ | −0.03 | −0.037 | −0.033 | −0.037 |
| (most deprived) | (−0.047 to −0.013) | (−0.056 to −0.017) | (−0.041 to −0.026) | (−0.050 to −0.025) |

Continued

**Table 5** Continued

| | First stage: time since last saw GP | | Second stage: EQ-5D utility score | |
|---|---|---|---|---|
| | **Female** | **Male** | **Female** | **Male** |
| | **(1)** | **(2)** | **(3)** | **(4)** |
| N | 570 626 | 662 258 | 570 626 | 662 258 |

Data source: GPPS pooled data for 2012–2017.
95% CIs in parentheses.
Models additionally include fixed effects for survey year and local authority district of residence.
*The reference category for age group is 16–24 years.
†The reference category for ethnic group is white.
‡The reference category for deprivation is decile 5.
GP, general practitioner; GPPS, General Practice Patient Survey.

that having an inflexible job represents an additional barrier to using routine healthcare, which may be one factor contributing to negative health effects of inflexible jobs.

The occupations which have seen increased home working during the pandemic (typically those where work can be carried out using information technology workarounds) are disproportionately those which are well paid[31] and require higher qualifications and more experience.[2] An increase in flexible working in these occupations could make already 'good' jobs even better, widening the gap between 'good' jobs and 'bad' jobs.[32] This could contribute to widening inequalities in health.

### Limitations

There are well-known empirical barriers to establishing a causal relationship between job characteristics and health. In particular, people with certain unobservable characteristics have a tendency to self-select into certain occupations. Part of the association we found between inflexible jobs and health is likely to be due to self-selection and/or reverse causality.

The people we identified as having inflexible jobs may also have been more likely to have jobs with other 'bad' characteristics such as a highly stressful work environment or a casual employment contract. We were unable to control for other job characteristics at an individual level, as this information is not included in the GPPS. Therefore, it is possible that the estimated effect of job inflexibility may be partly attributable to a correlated job characteristic. Data on employment composition are available at the LAD level; however, when we included indicators for the percentage share of each occupational classification in the regression models, they were jointly insignificant.

We identified people who had an inflexible job as those who responded that they were unable to go to see their GP in their usual working hours. There are, in fact, two possible types of job that would result in giving this response: those who worked 'normal hours' (eg, 09:00–17:00) but were unable to take time off during those hours to visit their GP; and those who worked unsociable

hours, so could not visit their GP during their working hours because the GP service was not open. Both of these job features are less desirable and may have a harmful effect on health.

The source of individual-level data we used is from a survey intended to gather information on patients' experience of general practice (GPPS). It is possible that people were more likely to respond to the survey if they had used their general practice recently. We used the survey weights provided with the GPPS to adjust for this, but it is still possible that this may affect generalisability to the general population.

### CONCLUSION

This is the first study to examine the relationship between job flexibility and health and healthcare utilisation in a large dataset. We found that having an inflexible job—defined as not being able to take time away from work during usual working hours to visit a medical professional—was associated with reduced HRQoL and less recent visits to see a GP or nurse. This relationship was consistent for both men and women.

Individuals living in more deprived areas were more likely to report having an inflexible job, and this inflexible job was more likely to have a negative impact on their health. This suggests that flexible working may have the potential to either exacerbate or reduce existing inequalities in health, depending on how it is distributed throughout the working population.

**Acknowledgements** We are grateful for comments and suggestions received from colleagues at HOPE and the Melbourne Institute: Applied Economic and Social Research. Additional thanks to Nordic Health Economists' Study Group 2019 conference attendees for a useful discussion of a previous version.

**Contributors** MS had the original idea for the study. MS, LAM and CM designed the analysis. CM carried out the analysis. CM, LAM and MS contributed substantially to the interpretation of results and the writing of the manuscript. CM is responsible for the overall content as the guarantor.

**Funding** This research was supported by the National Institute for Health Research (NIHR) Applied Research Collaboration Greater Manchester (NIHR200174). CM received funding from an NIHR Research Methods Fellowship

(RM-FI-2017-08-018). LAM was supported by the Medical Research Council, through a Skills Development Fellowship (MR/N015126/1). MS is an NIHR Senior Investigator.

**Disclaimer** The views expressed are those of the authors and not necessarily those of the MRC, the NIHR or the Department of Health and Social Care.

**Map disclaimer** The inclusion of any map (including the depiction of any boundaries therein), or of any geographic or locational reference, does not imply the expression of any opinion whatsoever on the part of BMJ concerning the legal status of any country, territory, jurisdiction or area or of its authorities. Any such expression remains solely that of the relevant source and is not endorsed by BMJ. Maps are provided without any warranty of any kind, either express or implied.

**Competing interests** None declared.

**Patient and public involvement** Patients and/or the public were not involved in the design, or conduct, or reporting, or dissemination plans of this research.

**Patient consent for publication** Not required.

**Ethics approval** Not applicable.

**Provenance and peer review** Not commissioned; externally peer reviewed.

**Data availability statement** Data may be obtained from a third party and are not publicly available. The General Practice Patient Survey data that support the findings of this study are available from NHS England but restrictions apply to the availability of these data, which were used under licence for the current study, and so are not publicly available.

**ORCID iDs**
Charlie Moss http://orcid.org/0000-0002-4694-378X
Luke Aaron Munford http://orcid.org/0000-0003-4540-6744

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
