## [Reviewer comments · BMJ Open]

ARTICLE DETAILS

TITLE (PROVISIONAL)	Associations between inflexible job conditions, health and healthcare utilisation in England: retrospective cross-sectional study
AUTHORS	Moss, Charlie; Munford, Luke; Sutton, Matt

VERSION 1 – REVIEW

REVIEWER	Alice Mannocci Sapienza University of Rome
REVIEW RETURNED	18-May-2022

GENERAL COMMENTS	Many thanks for the opportunity oreview the paper. It is well done, I don't have many comments. - I suggest to report the goodness of fit of the regression models in tables 3 and 4.- I agree with the presentation of the estimate the models for male and females separately, but I think that it is usefull to report the analisys with the gender too. Cosider the opportunity to present these results in supplementary files and report just the comments in the Results paragraph.
--

REVIEWER	Andrew Bryce The University of Sheffield, Economics
REVIEW RETURNED	09-Jun-2022

GENERAL COMMENTS	This paper uses data from the GPPS to estimate the association between inflexible work (defined as whether one can get time off during normal working hours to visit the GP) and health outcomes in England. I think this is an interesting and well-designed study with the potential to make a contribution to the work and health literature. A clear weakness of the paper is that it does not provide any evidence of causality. However, the authors are completely upfront about this and I find no reason to reject the paper on these grounds alone. The cross-sectional nature of the data makes it very difficult to make causal inference but I make some suggestions for additional analyses that could potentially strengthen the findings. 1. As it stands, the paper presents as its main results the direct association between inflexible work and EQ-5D scores (Tables 2 and 3). For me, I don't think these results are particularly interesting, partly because these associations already seem to be evident in the literature (although it is good to replicate and corroborate the findings of others) and partly because of the reverse causality and selection bias issues identified by the authors. To my mind, the real contribution of this paper lies in the mediating effect of healthcare utilisation and the paper should be framed in this way.
---

	2. The results presented in Table 4 are more interesting but raise a very important question which is not discussed at all. To what extent are visits to the GP or nurse classed as preventative as opposed to curative healthcare? In other words, if somebody does not visit their surgery for several years, does this constitute a poor health behaviour or is this a good outcome because it indicates that their health has been good and they have not required treatment? As such, I think there needs to be a much stronger theoretical framework outlining the three-way relationship between inflexible work, GP/nurse visits and health outcomes. Perhaps, rather than having separate models for the different outcomes, the authors may consider how they can integrate the GP/nurse visits variable into the main model, for example as a mediator between inflexible work and health. 3. Specifications 5 and 6 in Table 4 are also interesting but again the interpretation of the results is not straightforward because presumably only people with a specific health need will be trying to contact an out-of-hours GP. Moreover, we don't know whether the reason for contacting an out-of-hours service was due to not being able to attend a routine appointment in work time or due to having an acute health need requiring urgent treatment. We also don't know whether the patient was successful in securing an out-of-hours appointment. Assuming they were successful, there is an argument to suggest that an out-of-hours service might be a perfect substitute for seeing one's own GP, so patients may not be worse off by using an out-of-hours service. It may not be possible to unpack these questions from the data but some discussion of these issues would be very helpful. 4. In summary, the most interesting question that this paper should be answering is as follows: does inflexible work lead to unmet healthcare needs? The paper does not really answer this question as it stands and this is probably due to data limitations. However, I think the authors should explore options for identifying this effect. For example, is there an association between inflexible work and healthcare utilisation controlling for health (either indicators from EQ-5D or, if available, specific health conditions or shocks that would ideally require intervention from a GP or nurse)? Or is there spatial variation that could be exploited in some way? For example, are there any differential effects between rural and urban residents (where there is presumably a difference in access to GP services, particularly out-of-hours services, but maybe less of a difference in underlying health needs)? Can other data be used to identify LADs or LSOAs with a more restricted supply of primary healthcare where we would expect to see less flexibility in available appointment times? Minor comments 1. The EQ-5D measures are interpreted linearly despite being derived from ordinal questions. I think this is fine, if this is how it is usually treated in the literature, but as a robustness check I would like to see whether the results change in an ordinal specification. 2. Similarly, the choice to estimate the effects on GP/nurse visits as an interval regression seems appropriate. However, I have some concern about the "more than 12 months ago" category being interpreted as 12-24 months and the "never seen a GP" category
--	--

	being interpreted as 24+ months (Supplementary Table 1). I imagine that many of those who say “more than 12 months ago” last saw their GP/nurse much longer ago than 24 months (e.g. those who have lived in the area for a long time and have not had any reason to go to the surgery for several years). Again, I think that one or more robustness checks looking at different intervals (plus maybe an extensive margin type estimation looking at the difference between “never” and “ever”) should help to mitigate this concern. 3. I like the subgroup analysis but the paper does not currently report on whether the differences between subgroups are statistically significant. Using interactions rather than subgroups is one way to address this. 4. The authors control for local occupational composition but do not report these coefficients. I would like to see these coefficients (even if they are not reported in the paper itself). Are they significant? Do the main results change if these controls are removed? If not, I would like to see a stronger justification for their inclusion. The authors assert that these controls help to mitigate bias due to selection into different occupations. I am not convinced by this argument, so please can the authors provide a more detailed explanation of their thinking. 5. At the top of p16, the paper states that people may be more likely to respond to the survey if they have recently visited their surgery. The summary statistics suggest to me that this response bias may be an issue (0-3 months is the modal response for both GP and nurse). Is there any other data (e.g. on average frequency of GP visits among working age people) that could be used to corroborate this?
--	---

VERSION 1 – AUTHOR RESPONSE

Reviewer: 1

Dr. Alice Mannocci, Sapienza University of Rome Comments to the Author:

Many thanks for the opportunity oreview the paper.

It is well done, I don't have many comments.

- I suggest to report the goodness of fit of the regression models in tables 3 and 4.

We have added adjusted R^2 (or pseudo R^2 , as appropriate) to the regression results in Tables 2, 3 and 4. For the interval regressions, Stata does not provide an R^2 statistic, because it is not appropriate for this type of regression. We instead present AIC and BIC statistics for those regressions.

- I agree with the presentation of the estimate the models for male and females separately, but I think that it is usefull to report the analisys with the gender too. Cosider the opportunity to present these results in suplementry files and report just the comments in the Results paragraph.

We used the full sample to estimate the main model for both males and females and included an indicator for female gender, and interaction between female gender and having an inflexible job (Supplementary Table 6). We have amended the text to include the following in the results section:

“Respondents living in more deprived areas were more likely to have an inflexible job compared to those living in less deprived areas (Table 2; columns (1) and (2)). Older respondents were less likely than younger respondents to report having an inflexible job. *Women were more likely to have an inflexible job than men (Supplementary Table 6; column (1)).*”

“There was both a deprivation and age gradient on the effect of an inflexible job on health (Figure 2; *Supplementary Table 6*). *The interaction effects suggest that the negative effect of having an inflexible job on health was larger in magnitude for people living in a more deprived area compared to those living in a less deprived area; larger in magnitude for older people compared to younger people; and larger for women, compared to men.*”

Reviewer: 2

Dr. Andrew Bryce, The University of Sheffield Comments to the Author:

This paper uses data from the GPPS to estimate the association between inflexible work (defined as whether one can get time off during normal working hours to visit the GP) and health outcomes in England. I think this is an interesting and well-designed study with the potential to make a contribution to the work and health literature.

We are grateful for this encouragement.

A clear weakness of the paper is that it does not provide any evidence of causality. However, the authors are completely upfront about this and I find no reason to reject the paper on these grounds alone. The cross-sectional nature of the data makes it very difficult to make causal inference but I make some suggestions for additional analyses that could potentially strengthen the findings.

1. As it stands, the paper presents as its main results the direct association between inflexible work and EQ-5D scores (Tables 2 and 3). For me, I don't think these results are particularly interesting, partly because these associations already seem to be evident in the literature (although it is good to replicate and corroborate the findings of others) and partly because of the reverse causality and selection bias issues identified by the authors. To my mind, the real contribution of this paper lies in the mediating effect of healthcare utilisation and the paper should be framed in this way.

We believe there is a considerable contribution in the associations we present between having an inflexible job and health:

- 1) The EQ-5D is a rich, validated measure of health-related quality of life (HRQoL) used in a variety of settings. A scoring algorithm is available to enable it to be scored in terms of

population preferences for different aspects of health. The presence of the EQ-5D in the GPPS also allows us to examine the components of HRQoL separately using the five domains. The richness of the outcome measures exceeds any of the existing studies in the literature.

- 2) Our sample size is much larger than those used in previous studies meaning that we can estimate the association with more precision and we can analyse how the strength of the association varies across population groups.

As one of our bulletpoints in “strengths and limitations of this study” we state:

“We were able to use the EQ-5D-5L instrument which is a validated measure of health-related quality of life.”

We have also now amended our introduction to hopefully clarify the contribution we are making to the literature with our analysis of the relationship between work flexibility and health, and we have added more emphasis to the analysis of health care utilisation:

“However, previous studies suffer from several limitations. *Most likely due to insufficient sample size*, evidence of variation between population subgroups is lacking, even though we may expect some groups of employees to gain more from work flexibility (older employees and employees with other responsibilities).[24] Previous studies are also based predominantly on work-focused surveys that tend to include only crude measures of health and no information on healthcare utilisation.

We were unable to find a study of the effect of work flexibility on health care utilisation, despite this being a potentially important pathway contributing to the overall effect of work flexibility on health. For employees working “standard” daytime hours, those are the same hours that most routine health care provision is available. Although there may be some provision outside of “standard” hours, employees may be more likely to forego health care appointments if their job does not afford them the flexibility to attend during working hours.

We use large, repeated, health-focused cross-sectional surveys for England to consider full-time employed individuals who report their jobs are so inflexible that they cannot attend primary care appointments during working hours. We map the geographic distribution of inflexible jobs, estimate the associations between job flexibility and health and healthcare utilisation, and exploit the large sample for subgroup analyses by gender, age, and deprivation.”

We have also now extended our analysis, with a focus on the mediating effect of health care utilisation on health. We outline the approach we have taken below.

2. The results presented in Table 4 are more interesting but raise a very important question which is not discussed at all. To what extent are visits to the GP or nurse classed as preventative as opposed to curative healthcare? In other words, if somebody does not visit their surgery for several years, does this constitute a poor health behaviour or is this a good outcome because it indicates that their health has been good and they have not required treatment? As such, I think there needs to be a

much stronger theoretical framework outlining the three-way relationship between inflexible work, GP/nurse visits and health outcomes.

We have added to the introduction and methods sections to clarify how we are conceptualising the relationship between having an inflexible job, health and health care utilisation.

Introduction:

“We were unable to find a study of the effect of work flexibility on health care utilisation, despite this being a potentially important pathway contributing to the overall effect of work flexibility on health. For employees working “standard” daytime hours, those are the same hours that most routine health care provision is available. Although there may be some provision outside of “standard” hours, employees may be more likely to forego health care appointments if their job does not afford them the flexibility to attend during working hours.”

Methods section:

“The main analyses do not combine the associations of job inflexibility with health and with healthcare utilisation. As shown in Figure 1, it is possible that healthcare utilisation is one of the reasons health is adversely affected. If having an inflexible job acts as a barrier to accessing healthcare when it is needed, this would result in a negative effect on health care use. Because health care is expected to have a positive effect on health, this would create an additional negative effect of an inflexible job on health.

Figure 1: Graph of the hypothesised effect of an inflexible job on health

Perhaps, rather than having separate models for the different outcomes, the authors may consider how they can integrate the GP/nurse visits variable into the main model, for example as a mediator between inflexible work and health.

The cross-sectional nature of the data presents a major issue for estimating the effect of health care utilisation on health using a typical mediation analysis approach. We would expect the causal effect of utilisation on health to be positive. However, in a cross-sectional dataset such as the GPPS, the estimated association between health care utilisation and health would likely be negative due to reverse causality. This is the case in our dataset as, when we carry out a mediation analysis, we get the result that patients who have seen a GP or nurse more recently have lower HRQoL. For this reason, we do not believe that a mediation analysis is appropriate in this analysis and we have not included GP/nurse visits in the main model.

An alternative approach is to use two-stage least squares to decompose the effect of health care utilisation on health into that which is related to (1) having an inflexible job, and (2) all other factors that determine health care utilisation. In the first stage, we regress health care utilisation on having an inflexible job and in the second stage, we regress the EQ-5D utility score on predicted values of health care utilisation. This produces the expected results, in which having an inflexible job is associated with having seen a GP less recently, and having seen a GP less recently is associated with a reduction in the EQ-5D utility score.

We now present this additional analysis in the manuscript (new table, Table 5; and Supplementary Table 11) but we are very cautious about its validity. Amongst other things, for job inflexibility to be a valid instrument, it needs to be excludable from the health equation. The occupational health theories we have outlined in the introduction and our diagram of hypothesised effects in the Methods section suggest there are many channels besides that through health care utilisation through which job inflexibility can affect health. Moreover, as the system is just identified (one instrument and one endogenous variable), we cannot test the statistical validity of the over-identification restriction. We have highlighted these limitations, but we have added this analysis to the paper as we believe it remains a useful illustration of the proposed relationships between the outcomes we examine.

In the methods section we describe these analyses:

“To test this pathway, we used two-stage least squares (2SLS) to decompose the effect of health care utilisation on health into that which is related to (1) having an inflexible job, and (2) all other factors that determine health care utilisation. In the first stage, we estimated the association between the time since the respondent last saw a GP or nurse, and whether they have an inflexible job. In the second stage, we estimated the association between the predicted values for the time since the respondent last saw a GP/nurse and HRQoL. In these regressions, we treat the time since the respondent last saw a GP/nurse as a continuous variable ranging from one to five (Supplementary Table 1).”

And the results in the results section:

“The results of the second stage of the two-stage least squares regression show the effect of health care use on the EQ-5D utility score when healthcare use is predicted by job inflexibility (Table 5; Supplementary Table 12). A longer predicted duration of time since the individual had seen a GP (or nurse) was negatively associated with the EQ-5D utility score. Therefore, the element of healthcare use that is negatively associated with job inflexibility is also associated with lower health

3. Specifications 5 and 6 in Table 4 are also interesting but again the interpretation of the results is not straightforward because presumably only people with a specific health need will be trying to contact an out-of-hours GP. Moreover, we don't know whether the reason for contacting an out-of-hours service was due to not being able to attend a routine appointment in work time or due to having an acute health need requiring urgent treatment. We also don't know whether the patient was successful in securing an out-of-hours appointment. Assuming they were successful, there is an argument to suggest that an out-of-hours service might be a perfect substitute for seeing one's own GP, so patients may not be worse off by using an out-of-hours service. It may not be possible to unpack these questions from the data but some discussion of these issues would be very helpful.

It is difficult to unpack some of these issues from the available data. However, it is certainly the case that out-of-hours visits are more costly to the NHS than in-hours appointments because they are reimbursed as an additional service. Even if people with inflexible jobs only substitute in-hours care with out-of-hours care, this is more costly to the NHS.

We previously had a paragraph in the discussion about this general point, but we have amended it to include the specific point that out-of-hours appointments are more costly than regular appointments:

"Typically, the onus has been put on the health care system to accommodate inflexible jobs by offering "out-of-hours" appointments in the evenings and on weekends, *at greater cost to the NHS than regular GP appointments*. However, perhaps there should be a greater responsibility for employers to ensure that systems are in place so that their employees are able to attend routine health care appointments during usual working hours, when they need to."

4. In summary, the most interesting question that this paper should be answering is as follows: does inflexible work lead to unmet healthcare needs? The paper does not really answer this question as it stands and this is probably due to data limitations.

However, I think the authors should explore options for identifying this effect. For example, is there an association between inflexible work and healthcare utilisation controlling for health (either indicators from EQ-5D or, if available, specific health conditions or shocks that would ideally require intervention from a GP or nurse)?

Or is there spatial variation that could be exploited in some way? For example, are there any differential effects between rural and urban residents (where there is presumably a difference in access to GP services, particularly out-of-hours services, but maybe less of a difference in underlying health needs)? Can other data be used to identify LADs or LSOAs with a more restricted supply of primary healthcare where we would expect to see less flexibility in available appointment times?

We do not believe that the information relating to health care use in these data are detailed enough to get an accurate picture of unmet need. However, we have now included seventeen

indicators for specific long-term conditions as controls in the health care utilisation analyses to adjust for some identified indications of health care need.

Comparing urban to rural areas would be a good approach if the underlying health needs were similar. However, there is evidence that underlying health needs are not the same in different types of areas; health need tends to be greater in urban compared to rural areas. That said, a recent report by the Chief Medical Officer found that health needs and their underlying drivers are similar in coastal towns to urban areas, showing the complexity of geographical variations in health (<https://www.gov.uk/government/publications/chief-medical-officers-annual-report-2021-health-in-coastal-communities>).

Rather than use measures of primary care availability by geographical area, the GP Patient Survey allows us to look at indicators of appointment availability at the GP practice with which the individual is registered. We have now added a measure from the GPPS, which we included at the practice level to avoid potential issues relating to individual response bias. Specifically, we have controlled for the percentage of patients at each practice who reported “yes” to the question: “last time you wanted to see or speak to a nurse from your GP surgery, were you able to get an appointment to see or speak to someone?”.

The full regression results for the models with these new controls included are presented in Supplementary Tables 8 and 11. For all models, the coefficient on inflexible job changes slightly when the new controls are included, but the results are qualitatively the same as in the previous version of the manuscript.

Minor comments

1. The EQ-5D measures are interpreted linearly despite being derived from ordinal questions. I think this is fine, if this is how it is usually treated in the literature, but as a robustness check I would like to see whether the results change in an ordinal specification.

We now use ordered logit models to analyse the EQ-5D domains (Table 3). The results are qualitatively the same as the previous linear models.

2. Similarly, the choice to estimate the effects on GP/nurse visits as an interval regression seems appropriate. However, I have some concern about the “more than 12 months ago” category being interpreted as 12-24 months and the “never seen a GP” category being interpreted as 24+ months (Supplementary Table 1). I imagine that many of those who say “more than 12 months ago” last saw their GP/nurse much longer ago than 24 months (e.g. those who have lived in the area for a long time and have not had any reason to go to the surgery for several years). Again, I think that one

or more robustness checks looking at different intervals (plus maybe an extensive margin type estimation looking at the difference between “never” and “ever”) should help to mitigate this concern.

We have added three sensitivity analyses to the supplementary material (Supplementary Tables 9 and 10). In the first we use the interval 12 to 60 months for the response “more than 12 months ago” and 60+ months for “never seen a GP”. In the second we use the interval 12+ months for both of those responses. In the third, we use logistic regression to look only at the extensive margin.

In all cases, the results are qualitatively the same as the main analysis: having an inflexible job is associated with less recent use of GP/nurse.

3. I like the subgroup analysis but the paper does not currently report on whether the differences between subgroups are statistically significant. Using interactions rather than subgroups is one way to address this.

We have now estimated the effect of having an inflexible job on the EQ-5D utility score using the full sample, including interaction terms. We present the results in Supplementary Table 6 and the coefficients on the IMD and age interaction terms in Figure 2.

4. The authors control for local occupational composition but do not report these coefficients. I would like to see these coefficients (even if they are not reported in the paper itself). Are they significant? Do the main results change if these controls are removed? If not, I would like to see a stronger justification for their inclusion. The authors assert that these controls help to mitigate bias due to selection into different occupations. I am not convinced by this argument, so please can the authors provide a more detailed explanation of their thinking.

Our rationale for adjusting for the occupational composition was that it would provide some adjustment for other “bad” job characteristics that vary by occupation, rather than mitigate bias due to selection. However we have now tested the joint significance of the occupational composition indicators and they were insignificant, so we have removed them from the analysis.

We have amended the following paragraph in the limitations section:

“The people we identified as having inflexible jobs may also have been more likely to have jobs with other “bad” characteristics such as a highly stressful work environment or a casual employment contract. We were unable to control for other bad job characteristics at an individual level, as this information is not included in the GPPS. *Therefore, it is possible that the estimated effect of job inflexibility may be partly attributable to another correlated job characteristic. Data on employment composition are available at the LAD level, however when we included indicators for the percentage share of each occupational classification in the regression models they were jointly insignificant.*”

5. At the top of p16, the paper states that people may be more likely to respond to the survey if they have recently visited their surgery. The summary statistics suggest to me that this response bias

may be an issue (0-3 months is the modal response for both GP and nurse). Is there any other data (e.g. on average frequency of GP visits among working age people) that could be used to corroborate this?

To address this problem, we used the survey weights provided with the GPPS data. They are intended to adjust for several things including non-response. The descriptive statistics in the previous submission (Table 1) were not weighted, which was in error. (Note the survey weights were used in the regressions in the previous submission.) This has now been rectified. In the weighted summary statistics, the percentage of respondents who had seen their GP in the last 3 months is much lower than without the weights.

In the methods section we state:

“We estimated all models with robust standard errors, and used the weights provided with the survey data which correct for the GPPS sampling design and reduce the impact of non-response bias.”

And we have altered the point in the limitations section to state:

“The source of individual level data we used is from a survey intended to gather information on patients’ experience of general practice (GPPS). It is possible that people were more likely to respond to the survey if they had used their general practice recently. *We used the survey weights provided with the GPPS to adjust for this, but it is still possible that this may affect generalisability to the general population.*”